

# Altitudinal patterns in breeding bird species richness and density in relation to climate, habitat heterogeneity, and migration influence in a temperate montane forest (South Korea)

Jin-Yong Kim[1,2], Sanghun Lee[3], Man-Seok Shin[1], Chang-Hoon Lee[3], Changwan Seo[4] and Soo Hyung Eo[2]

[1] Division of Ecosystem Services and Research Planning, National Institute of Ecology, Seocheon, South Korea
[2] Department of Forest Resources, Kongju National University, Kongju, South Korea
[3] Division of Basic Ecology, National Institute of Ecology, Seocheon, South Korea
[4] Division of Ecological Survey Research, National Institute of Ecology, Seocheon, South Korea

Corresponding author
Soo Hyung Eo, eosh@kongju.ac.kr

## ABSTRACT

Altitudinal patterns in the population ecology of mountain bird species are useful for predicting species occurrence and behavior. Numerous hypotheses about the complex interactions among environmental factors have been proposed; however, these still remain controversial. This study aimed to identify the altitudinal patterns in breeding bird species richness or density and to test the hypotheses that climate, habitat heterogeneity (horizontal and vertical), and heterospecific attraction in a temperate forest, South Korea. We conducted a field survey of 142 plots at altitudes between 200 and 1,400 m a.s.l in the breeding season. A total of 2,771 individuals from 53 breeding bird species were recorded. Altitudinal patterns of species richness and density showed a hump-shaped pattern, indicating that the highest richness and density could be observed at moderate altitudes. Models constructed with 13 combinations of six variables demonstrated that species richness was positively correlated with vertical and horizontal habitat heterogeneity. Density was positively correlated with vertical, but not horizontal habitat heterogeneity, and negatively correlated with migratory bird ratio. No significant relationships were found between spring temperature and species richness or density. Therefore, the observed patterns in species richness support the hypothesis that habitat heterogeneity, rather than climate, is the main driver of species richness. Also, neither habitat heterogeneity nor climate hypotheses fully explains the observed patterns in density. However, vertical habitat heterogeneity does likely help explain observed patterns in density. The heterospecific attraction hypothesis did not apply to the distribution of birds along the altitudinal gradient. Appropriate management of vertical habitat heterogeneity, such as vegetation cover, should be maintained for the conservation of bird diversity in this area.

## INTRODUCTION

Altitudinal changes in bird species diversity provide important information on the limitation of species distribution within mountain areas (*Adolfo & Navarro, 1992*; *Kosicki, 2017*) and often serve as time-space substitutes and provide valuable predictive information (*Chamberlain et al., 2016*). For many decades, studies on distribution patterns along altitudinal gradients have been of interest to many researchers. Most commonly recognized pattern was decreasing richness with increasing elevation (*Terborgh, 1977*; *Stevens, 1992*; *Herzog, Kessler & Bach, 2005*). However, recent studies have described that bird diversity patterns may be more complex (*Poulsen & Lambert, 2000*; *Rahbek, 2005*; *McCain, 2009*). *McCain (2009)* suggested that, from the point of view of climate zones, four elevational richness patterns are represented. These are (1) decreasing, (2) low plateau, (3) low plateau with a mid-elevational peak, and (4) mid-elevational peak. To explain these altitudinal patterns, numerous hypotheses have been proposed (*Rahbek, 2005*; *Rahbek et al., 2007*; *McCain, 2009*; *Pan et al., 2016*).

These hypotheses generally fall into four main categories: climatic, spatial, evolutionary history, and biological hypothesis (*Pianka, 1966*; *Gaston, 2000*; *McCain, 2009*). Climatic hypotheses are based on the theory that species diversity is affected by conditions such as temperature, rainfall, productivity, humidity, and cloud cover (*McCain, 2009*). Spatial hypotheses suggest that the spatial extent of species distribution is reduced with increasing altitude, and thus, species diversity is simultaneously reduced (*Sanders & Rahbek, 2012*; *Pan et al., 2016*). Biological hypotheses include competition and habitat heterogeneity and complexity (*MacArthur & MacArthur, 1961*; *Terborgh, 1977*; *McCain, 2009*). Finally, evolutionary history hypotheses are linked to speciation rates, migration, extinction rates, and phylogenetic niche conservation (*Diamond, 1988*; *Lomolino, 2001*; *Allen, Brown & Gillooly, 2002*; *McCain, 2009*). Evolutionary history hypotheses are based on the assumption that speciation takes place most rapidly at low altitude, and extinction rate is highest at mountaintops (*McCain, 2009*) and also contained intra- and interspecific relationships such as migration and niche conservation.

Among the numerous hypotheses, climatic and biological hypotheses are the most widely supported (*Lee et al., 2004*; *McCain, 2009*; *Pan et al., 2016*). Climatic variables are considered to be the main driver of bird diversity (*McCain, 2009*), and temperature shows a distinct pattern that decreases with increasing altitude, which directly affects the physiological tolerance of birds (*Currie et al., 2004*; *Pan et al., 2016*) and indirectly affects birds by influencing vegetation and food resources. Therefore, the climatic hypothesis has been tested in many studies. However, many mechanistic models cannot fully explain the relationship between contemporary climate and species diversity (*Currie et al., 2004*; *Rahbek et al., 2007*). Therefore, alternative one involved in biological hypotheses have emerged, and the importance of habitat heterogeneity has been noted (*Rahbek et al., 2007*). Generally, habitat heterogeneity can positively influence bird species richness (*Hurlbert, 2004*; *Pan et al., 2016*); therefore, the

hypotheses have been receiving increased attention despite the difficulties in measurement and definition (*Pan et al., 2016*). Habitat heterogeneity hypothesis proposes that a greater variety of habitat types per unit area and a greater complexity of vegetation structure lead to increased diversity (*MacArthur & MacArthur, 1961*; *Pan et al., 2016*). However, most studies have been limited in scope by only employing horizontal factors, such as the variety of habitat types per unit area (*Pan et al., 2016*).

Although several environmental variables affect species diversity according to altitude, it is important to consider intra- and interspecific relationships. Migration in breeding season, one of the evolutionary hypotheses, is not only an alternative mechanism to explain birth and death, but also an important process in itself (*Dingle & Drake, 2007*). Therefore, the relationship between migrant and resident plays a particularly important role in breeding season. According to the heterospecific attraction hypothesis, migrants use residents as a cue to identify sites suitable for breeding because residents occupy higher-quality sites (*Mönkkönen et al., 1997*; *Mönkkönen & Forsman, 2002*). Therefore, increasing migration should positively affect species richness and density of a given site. However, to the best of our knowledge, the heterospecific attraction hypothesis has not yet been applied in advanced studies along an altitudinal gradient.

This study aimed to identify the altitudinal patterns in breeding bird species richness or density in a temperate montane forest, and we tested the hypotheses that (1) climate, (2) horizontal habitat heterogeneity, (3) vertical habitat heterogeneity, and (4) heterospecific attraction to explain the cause of such patterns. Further information of the each hypothesis is as follows; (1) lower temperature negatively affects species richness or density along altitude, (2) higher habitat diversity positively affects species richness or density along altitude, (3) greater structural complexity in vegetation positively affects species richness or density along altitude, (4) increasing species richness or density are influenced by inflow of migratory bird.

## MATERIALS AND METHODS

### Study area

This study was carried out in a forest in Jirisan National Park, the largest national park in South Korea with a total area of 481.022 km$^2$ (Fig. 1). All field surveys were conducted with the approval and access permits from the Korea National Park Service. The altitude in the park ranges from 110 to 1,915 m above sea level (a.s.l). The vegetation of the subalpine forest (up to 1,400 m a.s.l) is characterized by tree species such as *Betula ermanii*, *Malus baccata*, *Picea jezoensis*, *Pinus koraiensis*, *Abies koreana*, *Quercus mongolica*, *Q. serrata*, *Q. variabilis*, *Stewartia pseudocamellia*, *Pinus densiflora*, and *A. holophylla* (*Gwon et al., 2013*). The study focused on montane forest areas between altitudes of 200 and 1,400 m a.s.l, because the altitudes above 1,500 m include ridges, most of which are populated by coniferous shrubs.
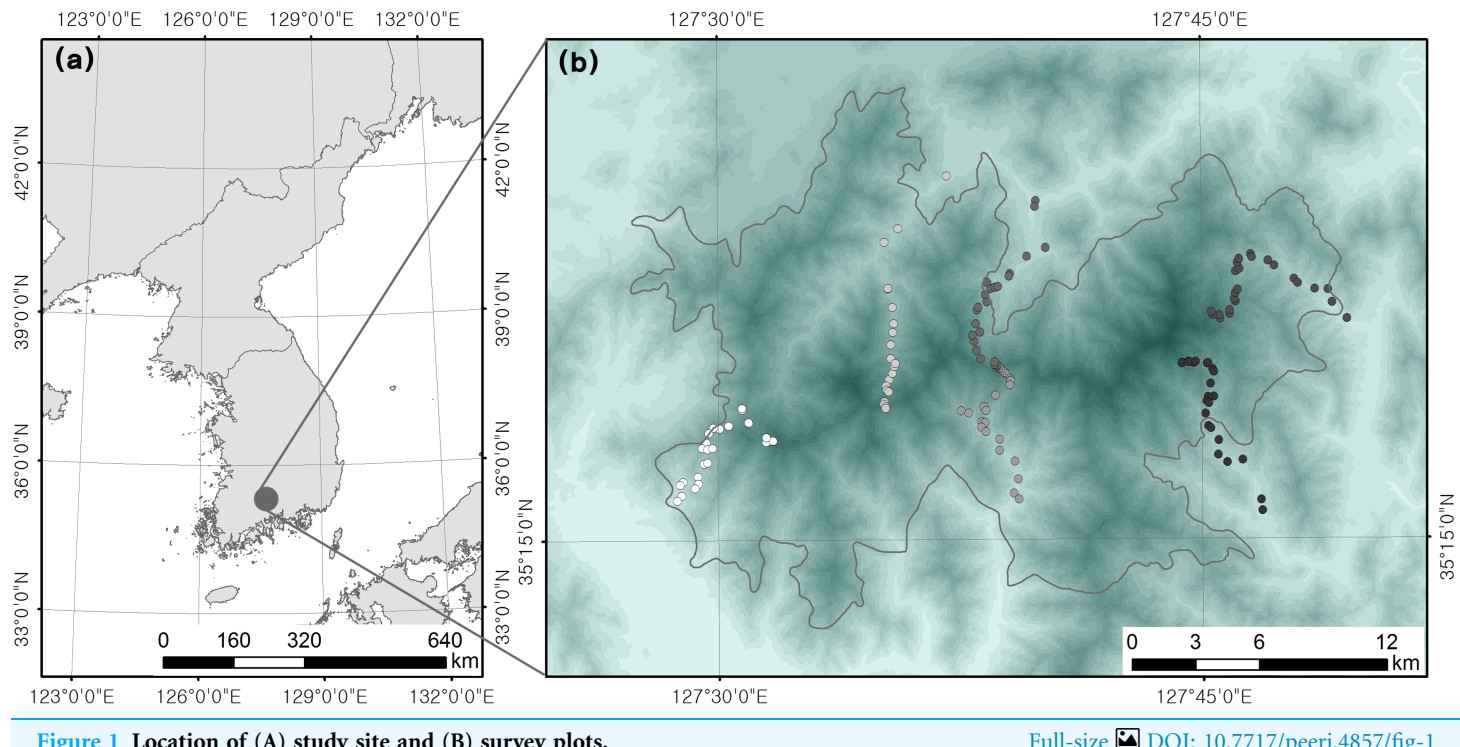

**Figure 1 Location of (A) study site and (B) survey plots.**

## Bird survey

A total of 142 plots were surveyed along the elevational gradient in mixed or deciduous forested areas, with coniferous forests excluded from the survey area to minimize the differences in bird communities according to forest type (Table S1). We randomly chose 10–12 plots within each 100 m elevation bracket within an altitudinal range of 200–1,400 m. The location of each plot was recorded using a Global Positioning System (GPS; Oregon 300; Garmin, Lenexa, KS, USA). Surveys of bird fauna and vertical coverage of vegetation were undertaken in every plot. Point counts of birds (*Reynolds, Scott & Nussbaum, 1980*) were carried out between late May and June 2015 to account for summer migratory arrivals. At each plot, all breeding bird seen and heard within a 50 m radius (0.8 ha) were recorded the No. of individuals and species using 15 min count period. Point count commenced directly after sunrise and continued until 8 a.m. in good weather conditions (without precipitation, fog, and prevalent wind). We did not count chicks, to reduce the change in the number of individuals caused by fledging of chicks. Nonbreeding species, which were classified as passing migrants, were eliminated from the analysis (Table S2).

## Climatic hypothesis variables (temperature and humidity)

We used the Weather Research and Forecasting (WRF) version 3.6 model to retrieve climate parameters, including mean spring temperature and relative humidity, on regional and local scales. These parameters were compiled over a three-month period using terrestrial data from the National Center for Environmental Prediction (NCEP)

Final (FNL) Operational Global Analysis data. Using these data, climate simulation with WRF was executed for April, May, and June 2015 at time intervals ($\Delta t$) of 180 s. Since the NCEP input data resolution of 1° is very coarse for regional or local climate simulations, the domains in this study were downscaled into two-way quadruples of 27, 9, 3, and 1 km with 31 vertical levels in WRF. Simulation outputs were produced every hour with a cumulus parameterization scheme by Kain and Fritsch (*Kain & Fritsch, 1993*), the WRF Double Moment 6-Class Microphysics Scheme (WSM6) (*Hong et al., 2010*) to simulate cloud physical processes, and the Yonsei University (YSU) PBL scheme (*Lee et al., 2011*) to parameterize turbulence in the planetary boundary layer. After simulation, habitat temperatures were extracted based on coordinates.

### Biological hypothesis variables (vertical and horizontal habitat heterogeneity)

To quantify vertical habitat heterogeneity, we surveyed the vertical coverage of vegetation at each sampling plot within 5 m radii. Within these circles, we classified vertical layers into understory (<2 m), midstory (2–10 m), and overstory (>10 m) vegetation. Coverage was classified into the following four categories: 0 (0% coverage), 1 (1–33% coverage), 2 (34–66% coverage), and 3 (67–100% coverage) (*Lee et al., 2011*; *Rhim, 2012*).

For horizontal habitat heterogeneity, we calculated the Shannon–Wiener diversity index (H′) using the area of that particular habitat type (abundance) and number of different habitat types (richness) (*Turner & Gardner, 2015*; *Pan et al., 2016*). The area and number of habitat types were extracted from land cover maps (Ministry of Environment, Republic of Korea) within a 150 m radius circle at each plot using ArcGIS 10.3 (ESRI, Redlands, CA, USA). The top categories of habitat type comprised anthropogenic, agricultural, managed and natural forestry, herbaceous, wetland, barren, and water areas. A total of 15 habitat types of sub categories (residential, commercial, roads, public facilities, rice paddy, farm land, orchard, deciduous, coniferous, mixed forest, natural grassland, artificial grassland, swamp, barren, water; Fig. 2) were defined and used for the habitat diversity index.

### Evolutionary hypothesis variable (migratory bird ratio)

To identify migration influence, we simply used the migratory bird ratio, which was calculated based on the ratio of the total number of species or individuals and the number of migratory species or individuals in each plot (*Helle & Fuller, 1988*; *Newton & Dale, 1996*). All birds detected were classified as residents or summer migrants. Migrants were defined as wintering in the tropical region of Southeast Asia and migrating to the study area for breeding purposes. Twenty-three species were identified as summer migrants and 30 species were defined as residents (Table S2).

### Data analyses

To investigate the distribution patterns of breeding bird species richness and individuals along an altitudinal gradient, we used the curve estimation function in SPSS 20. Best-fit curves (linear, quadratic, and exponential) were selected according to the

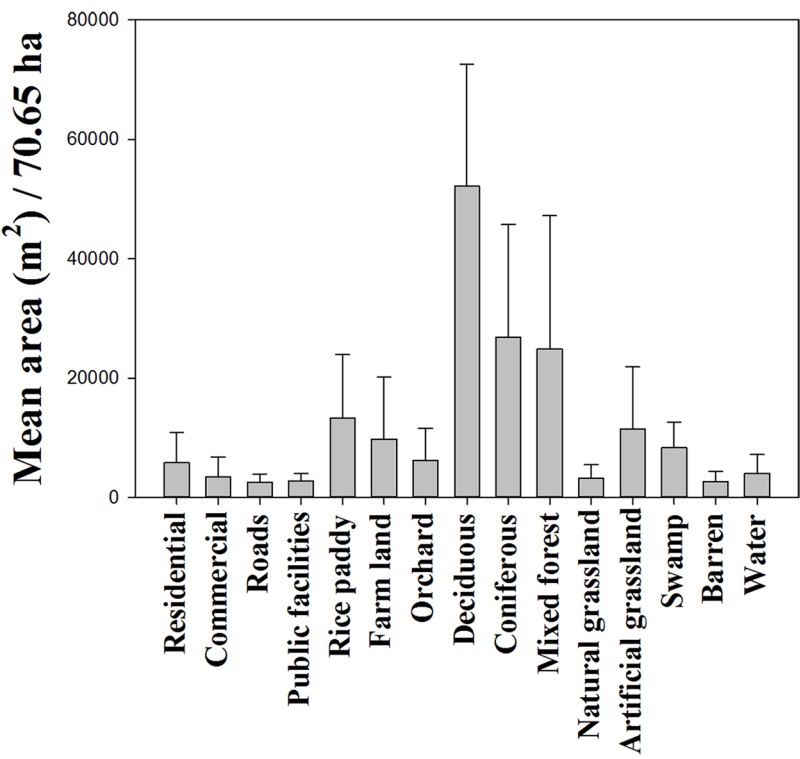

**Figure 2 Mean area of 15 habitat types within a 150m radius circle in study site.**

highest $R^2$ and significant *p*-values. Using the same method, we verified a linear relationship between the dependent variables (species richness and density) and independent variables (spring temperature, vertical coverage of vegetation, horizontal habitat diversity, and migratory bird ratio). The variables were surveyed and extracted from the same plot point, however have different spatial and temporal resolution. We set the longest temporal range to breeding season for which spring temperature was calculated, and bird and vegetation survey were investigated within least time to reduce a variance. The widest spatial range was set in horizontal habitat range, in which bird and vegetation survey were investigated.

We used model selection and multimodel inference using a generalized linear model (GLM). We developed a set of 13 candidate models using this GLM, using 13 combinations of variables to identify the causes of altitudinal patterns in bird species richness and density in relation to spring temperature, migratory bird ratio, vertical coverage of vegetation, and horizontal habitat diversity variables. Before adding variables to the model selection, we eliminated correlated predictors ($r \geq |0.7|$) with another variable. Once the models were created, we used information-theoretic methods (*Burnham & Anderson, 2002*) to choose from among the competing models by converting log-likelihood values calculated using Akaike's information criterion adjusted for small sample sizes (AICc) (*Akaike, 1974*) and Akaike weights ($w_i$). If we identified models with uninformative parameters, the parameters were eliminated from the model (*Arnold, 2010*).

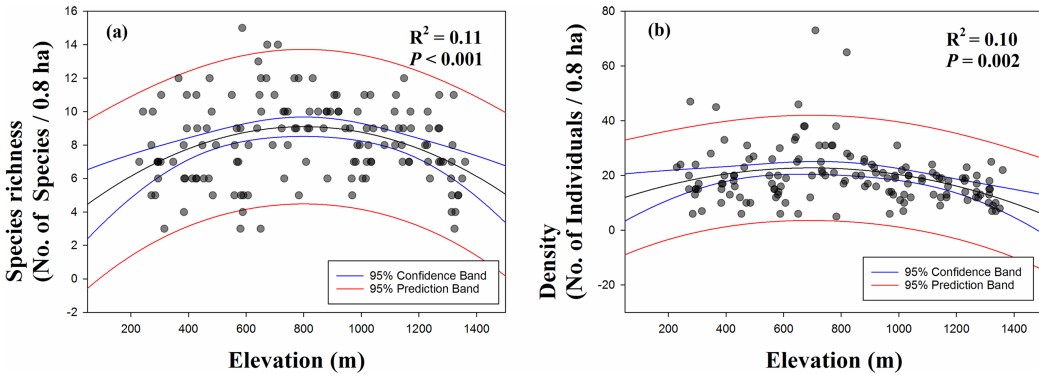

**Figure 3** **Distribution patterns of (A) species richness and (B) density along an altitudinal gradient.** Best-fit curves (linear, quadratic, and exponential) were selected according to the highest $R^2$ and significant *p*-values.

And then we reconstructed models without uninformative parameters. The high-confidence set of candidate models consisted of models with Akaike weights within 10% of the highest (*Royall, 1997*; *Lepczyk et al., 2008*), and these were used to compute model-averaged parameter estimates (*Burnham, Anderson & Huyvaert, 2011*). All statistical analyses were performed using R 3.3.2 (packages bbmle, AICcmodavg, and MuMin).

## RESULTS

### Altitudinal patterns in species richness and density

Fifty-three species were observed in the 142 survey plots during the breeding period surveyed, with a total of 2,771 individual birds. To verify the altitudinal patterns in species richness and numbers of individuals, we estimated best-fit curves. Breeding bird species richness showed a hump-shaped pattern along an altitudinal gradient ($R^2 = 0.11$, $p < 0.001$; Fig. 3A). A linear pattern of species richness was not significant in relation to altitude ($R^2 = 0.00$, $p = 0.820$). In addition, density showed a hump-shaped pattern ($R^2 = 0.10$, $p = 0.002$; Fig. 3B), rather than a linear pattern ($R^2 = 0.04$, $p = 0.019$).

### Relationships of species richness and density with different variables
#### *Single variable patterns*

Pearson's correlation analysis of nine environmental variables showed that spring temperature and relative humidity were highly correlated ($r = -0.951$; Table S3). Elevation showed strong correlations with spring temperature and relative humidity ($r = -0.977$, $r = 0.938$, respectively; Table S3). Although migratory ratio of species and individuals were correlated ($r = 0.851$; Table S3), these were not included in the same model. Therefore, elevation and relative humidity variables were eliminated from the curve estimation and model construction.

In the best-fit curve estimation between species richness, density, and environmental variables, species richness showed significant correlations with spring temperature ($R^2 = 0.08$, $p = 0.003$; Fig. 4A) and migratory bird ratio ($R^2 = 0.11$, $p < 0.001$; Fig. 4B), and were represented by hump-shaped curves. No relationships were observed between species

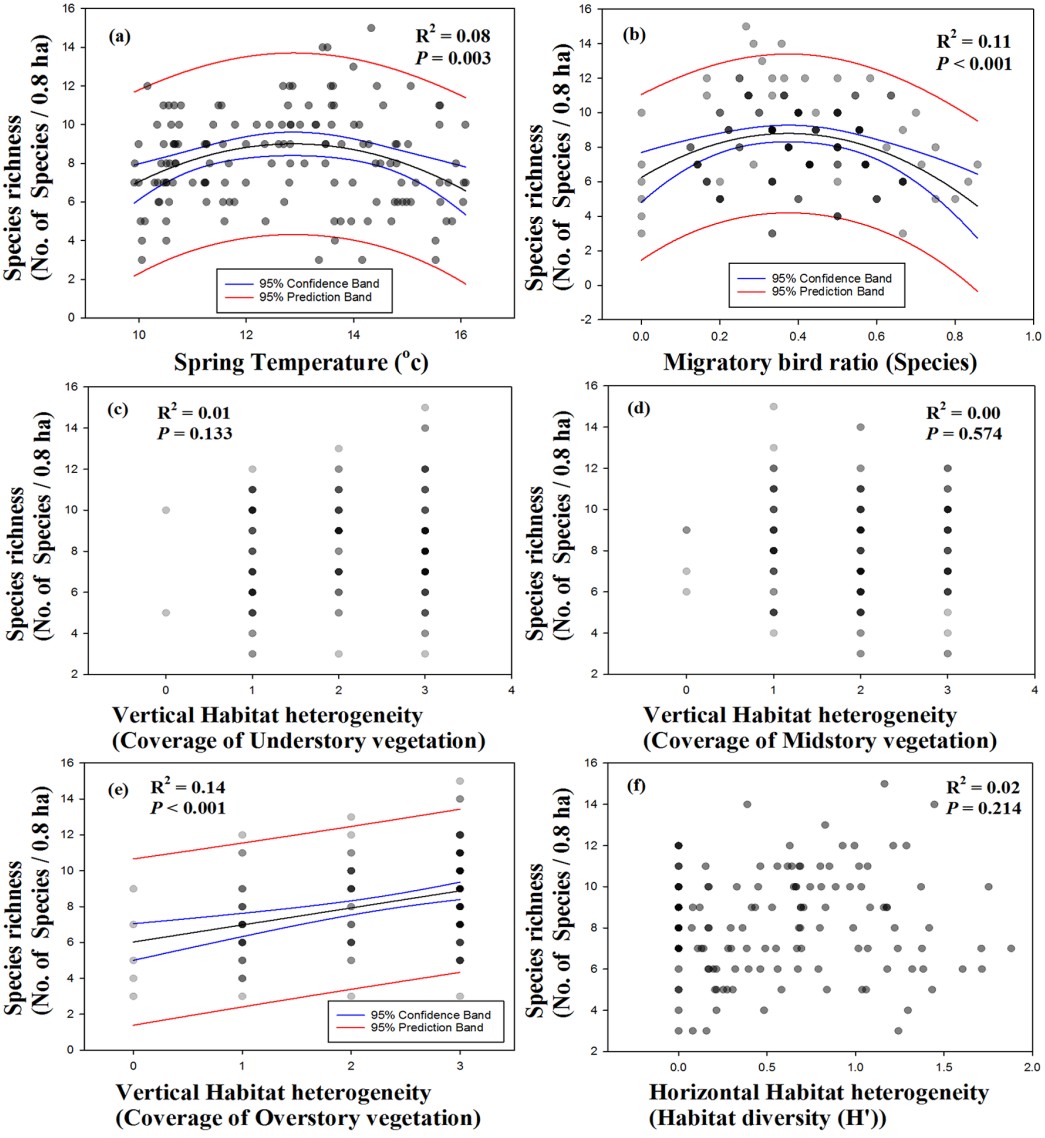

**Figure 4 Single variable patterns using best-fit curve function between species richness and variables.** Variables were consisted with (A) spring temperature, (B) migratory bird ratio, vertical ((C) under, (D) mid, (E) overstory vegetation), and (F) horizontal (habitat diversity) habitat heterogeneity.

richness and coverage of understory vegetation, midstory vegetation, or habitat diversity (Figs. 4C, 4D and 4F). Species richness and coverage of overstory vegetation showed a significant positive correlation ($R^2 = 0.14$, $p < 0.001$; Fig. 4E). Moreover, density showed a significant correlation with spring temperature in a hump-shaped pattern ($R^2 = 0.11$, $p < 0.001$; Fig. 5A). A decreasing pattern was observed between density and migratory bird ratio ($R^2 = 0.07$, $p = 0.006$; Fig. 5B), and coverage of under- and overstory vegetation represented a monotonically increasing pattern with increasing density ($R^2 = 0.03$, $p = 0.027$; $R^2 = 0.40$, $p = 0.017$; Figs. 5C and 5E). Other variables, including coverage of midstory vegetation and habitat diversity, did not show any significant correlations.
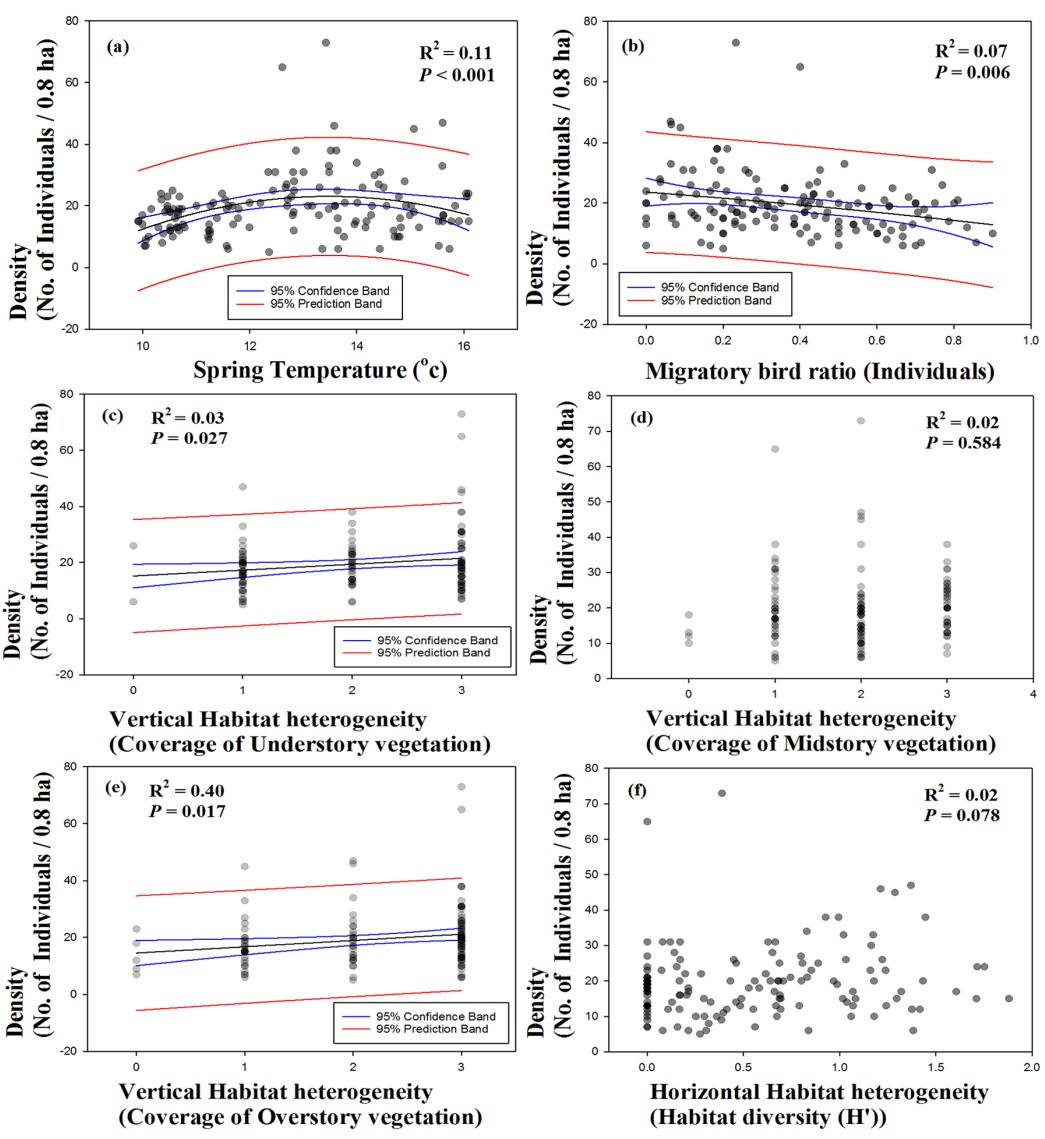

**Figure 5 Single variable patterns using best-fit curve function between density and variables.** Variables were consisted with (A) spring temperature, (B) migratory bird ratio, vertical ((C) under, (D) mid, (E) overstory vegetation), and (F) horizontal (habitat diversity) habitat heterogeneity.

## *Model selection and multimodel inference*

The set of candidate models with 13 combinations of six variables showed six models supported for species richness (Table 1). The best predictors of species richness were overstory vegetation, midstory vegetation, understory vegetation, habitat diversity, and migratory bird ratio ($w_i = 0.364$). Vertical coverage variables were included in all supported species richness models. A model including habitat diversity was 2.2 times more likely to explain species richness better than models excluding it ($w_i = 0.364$ vs. $w_i = 0.164$; Table 1). The Akaike weight was 1.8 times higher the inclusion of migratory bird ratio than when these parameters were excluded ($w_i = 0.364$ vs. $w_i = 0.197$; Table 1).
**Table 1** Model selection for predicting species richness according to spring temperature, migratory bird ratio, vertical (under, mid, overstory vegetation), and horizontal (habitat diversity) habitat heterogeneity.

| Response variables | Candidate models | AICc | ΔAICc | df | $w_i$ |
|---|---|---|---|---|---|
| Species richness (No. of species/0.8 ha) | (Best model) intercept + OV + MV + UV + HD + MRs | 637.7 | 0.0 | 7 | 0.364 |
| | Intercept + OV + MV + UV + HD | 638.9 | 1.2 | 6 | 0.197 |
| | Intercept + OV + MV + UV + MRs | 639.3 | 1.6 | 6 | 0.164 |
| | (Full model) intercept + OV + MV + UV + HD + ST + MRS | 639.5 | 1.8 | 8 | 0.149 |
| | Intercept + OV + MV + UV + HD + ST | 641.1 | 3.4 | 7 | 0.067 |
| | Intercept + OV + MV + UV + ST | 642.3 | 4.6 | 6 | 0.037 |
| | Intercept + OV + MV + UV | 643.4 | 5.6 | 5 | 0.022 |
| | Intercept + HD | 660.6 | 22.9 | 3 | <0.001 |
| | Intercept + MRs | 660.9 | 23.2 | 3 | <0.001 |
| | Intercept + ST | 661.5 | 23.8 | 3 | <0.001 |
| | Intercept + HD + ST | 662.4 | 24.6 | 4 | <0.001 |
| | Intercept + HD + MRs | 662.5 | 24.8 | 4 | <0.001 |
| | Intercept + ST + MRs | 662.9 | 25.2 | 4 | <0.001 |

Notes:
Coverage of UV, understory vegetation; MV, midstory vegetation; OV, overstory vegetation; ST, spring temperature; HD, habitat diversity; MRs, migratory bird ratio (species).

**Table 2** Model selection results for predicting species richness according to migratory bird ratio, vertical (under, mid, overstory vegetation), and horizontal (habitat diversity) habitat heterogeneity. Uninformative parameters (spring temperature) were removed from models.

| Response variables | Candidate models | AICc | ΔAICc | df | $w_i$ |
|---|---|---|---|---|---|
| Species richness (No. of species/0.8 ha) | (Best model, full model) intercept + OV + MV + UV + HD + MRs | 637.7 | 0.0 | 7 | 0.488 |
| | Intercept + OV + MV + UV + HD | 638.9 | 1.2 | 6 | 0.264 |
| | Intercept + OV + MV + UV + MRs | 639.3 | 1.6 | 6 | 0.219 |
| | Intercept + OV + MV + UV | 643.4 | 5.6 | 5 | 0.029 |
| | Intercept + HD | 660.6 | 22.9 | 3 | <0.001 |
| | Intercept + MRs | 660.9 | 23.2 | 3 | <0.001 |
| | Intercept + HD + MRs | 662.5 | 24.8 | 4 | <0.001 |

Notes:
Coverage of UV, understory vegetation; MV, midstory vegetation; OV, overstory vegetation; ST, spring temperature; HD, habitat diversity; MRs, migratory bird ratio (species).

When spring temperature was excluded in the species model, the Akaike weight was 2.4 times higher ($w_i$ = 0.364 vs. $w_i$ = 0.149; Table 1). Therefore, we regarded spring temperature as an uninformative parameter and excluded it in the next model and constructed seven models again (Table 2). As a results of seven models, the best supported model was full model ($w_i$ = 0.488; Table 2). The Akaike weight was 1.8 and 2.2 times higher the inclusion of migratory bird ratio and habitat diversity, respectively, than these parameters were eliminated from model ($w_i$ = 0.488 vs. $w_i$ = 0.264, $w_i$ = 0.488 vs. $w_i$ = 0.219; Table 2).

The results of model selection for predicting density showed three supported models (Table 3). The best model for predicting density included overstory vegetation, midstory vegetation, understory vegetation, habitat diversity, and migratory bird ratio (individuals)

**Table 3 Model selection for predicting density according to spring temperature, migratory bird ratio, vertical (under, mid, overstory vegetation), and horizontal (habitat diversity) habitat heterogeneity.**

| Response variables | Candidate models | AICc | ΔAICc | df | $w_i$ |
|---|---|---|---|---|---|
| **Density (No. of individuals/0.8 ha)** | (Best model) intercept + OV + MV + UV + HD + MRi | 1037.4 | 0.0 | 7 | 0.342 |
| | (Full model) intercept + OV + MV + UV + HD + ST + MRi | 1037.5 | 0.1 | 8 | 0.321 |
| | Intercept + OV + MV + UV + MRi | 1037.6 | 0.2 | 6 | 0.303 |
| | Intercept + OV + MV + UV + ST | 1042.6 | 5.3 | 6 | 0.025 |
| | Intercept + OV + MV + UV + HD + ST | 1044.8 | 7.4 | 7 | 0.008 |
| | Intercept + OV + MV + UV + HD | 1052.2 | 14.8 | 6 | <0.001 |
| | Intercept + MRi | 1055.1 | 17.8 | 3 | <0.001 |
| | Intercept + ST + MRi | 1056.7 | 19.3 | 4 | <0.001 |
| | Intercept + HD + MRi | 1056.7 | 19.4 | 4 | <0.001 |
| | Intercept + OV + MV + UV | 1057.8 | 20.5 | 5 | <0.001 |
| | Intercept + ST | 1059.4 | 22.0 | 3 | <0.001 |
| | Intercept + HD + ST | 1061.5 | 24.1 | 4 | <0.001 |
| | Intercept + HD | 1062.3 | 24.9 | 3 | <0.001 |

Notes:
Coverage of UV, understory vegetation; MV, midstory vegetation; OV, overstory vegetation; ST, spring temperature; HD, habitat diversity; MRi, migratory bird ratio (individuals).

($w_i$ = 0.342; Table 3). Vertical coverage variables and migratory bird ratio were included in all supported models. When habitat diversity was included in the density model, the Akaike weight was 1.13 times higher than when habitat diversity was eliminated from the model ($w_i$ = 0.342 vs. $w_i$ = 0.303; Table 3), and 1.07 times higher in the absence of spring temperature ($w_i$ = 0.342 vs. $w_i$ = 0.321; Table 3).

Multimodel-averaged parameter estimates of species richness, including the three supported models, showed positive correlations with overstory vegetation, understory vegetation, and habitat diversity ($p < 0.001$, $p = 0.025$, $p = 0.040$, respectively; Table 4). Density including the three supported models showed positive correlations with overstory vegetation and understory vegetation ($p < 0.001$, $p < 0.001$; Table 4) and a negative correlation with migratory bird ratio ($p < 0.001$; Table 4).

# DISCUSSION

## Altitudinal patterns of species richness and density

Altitudinal patterns in breeding bird species richness and density showed a hump-shaped pattern (Fig. 3), as found in previous studies (*Poulsen & Lambert, 2000*; *Lomolino, 2001*; *Ding et al., 2005*; *Pan et al., 2016*). Four main altitudinal patterns of species richness have been identified for geographical features such as climate type (tropical, subtropical, arid, and temperate), latitude, longitude, landmass type (islands and continents), altitude, and spatial scale (local and regional), but no relationships between elevational species richness and altitude or latitude have been observed (*McCain, 2009*). Most of the previously studied areas, which demonstrated a mid-peak pattern of species richness, were located in the northern and eastern regions of Asia and consisted of mountain in

**Table 4 Results of AICc-based multimodel inference of species richness and density.** Candidate models included those with Akaike weight within 10% of the highest value. Spring temperature, migratory bird ratio, vertical (under, mid, overstory vegetation), and horizontal (habitat diversity) habitat heterogeneity were used as an independent variables.

| Parameter | Model-averaged estimates | SE | *p*-Value | Importance value |
|---|---|---|---|---|
| **Species richness** | | | | |
| Intercept | 5.159 | 1.061 | <0.001*** | – |
| Understory vegetation | 0.499 | 0.220 | **0.025*** | 1.00 |
| Midstory vegetation | −0.124 | 0.233 | 0.597 | 1.00 |
| Overstory vegetation | 1.119 | 0.217 | <0.001*** | 1.00 |
| Habitat diversity | 0.862 | 0.416 | 0.040* | 0.77 |
| Migratory bird ratio (species) | −2.284 | 1.168 | 0.053 | 0.73 |
| **Density** | | | | |
| Intercept | 5.671 | 9.050 | 0.533 | – |
| Understory vegetation | 3.410 | 0.920 | <0.001*** | 1.00 |
| Midstory vegetation | 0.173 | 0.946 | 0.856 | 1.00 |
| Overstory vegetation | 3.340 | 0.876 | <0.001*** | 1.00 |
| Migratory bird ratio (individuals) | −15.134 | 4.236 | <0.001*** | 1.00 |
| Habitat diversity | 1.732 | 2.065 | 0.405 | 0.69 |
| Spring temperature | 0.927 | 0.650 | 0.158 | 0.33 |

**Notes:**
SE, standard error.
* $p < 0.05$.
** $p < 0.01$
*** $p < 0.001$.

arid zones (*Ding et al., 2005*; *McCain, 2009*; *Acharya et al., 2011*; *Pan et al., 2016*). Moreover, a study conducted in Eastern Himalaya showed a hump-shaped pattern (*Ding et al., 2005*). In contrast, studies from Southeast Asia have shown predominantly decreasing patterns (*McCain, 2009*). Best-fits to the null model generally showed more mid-peaks in local-scale studies than in regional studies (*McCain, 2009*). Our species richness pattern, found in East Asia and at a local scale, followed the most frequently identified hump-shaped pattern. Further, we identified a hump-shaped pattern of density. Density has been referred to in standardized methods but has not been reported in many studies (*Lomolino, 2001*) compared to species richness.

## Relationships of species richness and density with different variables

Previous studies have shown that vegetation cover and habitat diversity have strong positive relationships with species richness (*MacArthur & MacArthur, 1961*; *Hurlbert & Haskell, 2003*; *Hurlbert, 2004*; *Acharya et al., 2011*; *Pan et al., 2016*). Dense vertical vegetation coverage may play an important role in providing birds with breeding space, shelter, and food resources such as insects, which could contribute positively to bird species richness and density. According to the more-individuals hypothesis, density and energy use in communities is positively correlated with energy availability, and species richness can contribute to density (*Goetz et al., 2007*). Accordingly, our results showed

increasing species richness and density with increasing vertical overstory and understory vegetation cover (Table 4), and species richness showed a significant positive relationship with density (Fig. S1). Further, the present study demonstrated that species richness was affected by horizontal habitat diversity, but density was not (Table 4). High habitat diversity can increase species richness due to niche partitioning and providing habitat edges (*Best, Whitmore & Booth, 1990*), but high habitat diversity does not necessarily indicate high habitat quality with ample food resources. Therefore, the lack of a relationship between density and habitat diversity in this study might be because density increased with productivity and habitat quality (*Hurlbert, 2004*; *Goetz et al., 2007*). According to the habitat heterogeneity hypothesis (*MacArthur & MacArthur, 1961*; *Pan et al., 2016*), greater structural complexity in vegetation and more habitat types likely contributed to species richness in the present study. However, a larger number of habitat types did not influence the density.

We observed a negative relationship between density and migratory bird ratio, and no relationship was observed between species richness and migratory bird ratio (Fig. 5B; Table 4). Based on the heterospecific attraction hypothesis (*Mönkkönen et al., 1997*; *Mönkkönen & Forsman, 2002*), we predicted that the migratory bird ratio would have a positive effect on migrant species richness and density, and that the migratory bird ratio would increase with resident species richness and density. However, in the present study, a reduction in the migratory bird ratio led to an increase in density. Additionally, migrant species richness and density showed an increasing pattern along the altitude gradient, whereas resident species richness and density showed a mid-peak pattern along the altitude gradient (Fig. S2). It is unlikely that the migrants could choose a mid-elevation with higher vegetation coverage than the residents could (Fig. S2). Migrant species and individuals did not positively influence species richness and density, and they were not attracted to resident species. Therefore, the heterospecific attraction hypothesis was not applicable along the altitude gradient surveyed in the present study.

No relationships were found between species richness or density and climatic factors (Table 4) and a decreasing pattern of spring temperature along the altitudinal gradient was identified (Fig. S2). Numerous studies have shown a positive relationship between temperature and diversity (*McCain, 2009*). However, significantly stronger relationships between temperature and diversity can be found in humid mountain habitats than in dry mountain habitats (*McCain, 2009*). Furthermore, a negative relationship between density and climatic factors was found in a study conducted in Asia (*Pan et al., 2016*). Despite numerous studies on this phenomenon, the pattern has not been adequately explained (*Currie et al., 2004*; *Rahbek et al., 2007*). Most studies used the average annual temperature from the WorldClim database and conducted bird surveys across all seasons using considerably larger datasets that have constrained accuracy due to the sampling effort involved (*Lomolino, 2001*; *Ding et al., 2005*). However, in the present study, we used spring temperature values derived for micro-scale studies, and focused on breeding bird survey on a local scale in a short period in mixed and deciduous forest areas; this approach may have led to the variation in the findings. Another possible explanation is

that birds are restricted more by habitat quality for chick rearing than by temperature during the breeding season.

A single variable analysis showed no significant relationships between species richness and understory vegetation or habitat diversity (Figs. 4C and 4F); however, a significant relationship was observed in the modeling approach. Additionally, no differences in density were observed either in the single variable or in the modeling approach. Null hypothesis testing, similar to a simple linear correlation, has been used in many ecological studies and is currently being used in many areas. However, almost ecological phenomena has been often represented by nonlinear and multiple interaction among variables (*Landis et al., 2013*). For example, species should live at the proper temperature for the optimal thermal fitness during breeding season. But if there is no proper nesting resources, food and shelter, the species should choose a different habitat even the proper temperature for breeding. Consequentially, each variables does not affect the dependent variable, but multiple interactions of the variables. Therefore, alternative modeling approach is required for ecological studies and considered to be a more reliable method that avoiding uninformative, logical deficiencies and common misinterpretations of null hypothesis testing (*Anderson, Burnham & Thompson, 2000*; *Mönkkönen, Forsman & Bokma, 2006*). In order to understand the complex ecological phenomena, the use of multimodels is more reasonable and needs more efforts to clarify the relationship of the causative variables.

## CONCLUSION

Trends in species richness showed hump-shaped patterns along altitudinal gradients and were related to vertical vegetation coverage and horizontal habitat diversity. In addition, trends in density also showed hump-shaped patterns, with density related to vertical vegetation coverage and migratory bird ratio, but not to habitat diversity. No significant relationships were found between spring temperature and species richness or density. The results on species richness support the habitat heterogeneity hypothesis rather than the climate hypothesis, whereas those of species density do not support fully either hypothesis, and they were related to species richness and vertical vegetation coverage. The heterospecific attraction hypothesis was not applicable to the distribution of birds along the altitudinal gradient studied. Taken together, our findings indicate that management of vegetation cover would be an appropriate strategy for avian conservation in this region. To achieve a better understanding of the specific reasons for the distribution of birds along altitudinal gradients, further studies on the interactions among species related to niche and competition are required.

### Funding

This study was funded by a grant from the Kongju National University in 2013 and the project (NIE-BR-2018-11; NIE-BR-2015-12; 2014001310009) from the National Institute of Ecology (Misnistry of environment) of the Republic of Korea. The funders had no role in study design, data collection and analysis, decision to publish, or preparation of the manuscript.

## Grant Disclosures

The following grant information was disclosed by the authors:
Kongju National University.
National Institute of Ecology (Ministry of environment) of the Republic of Korea: NIE-BR-2018-11; NIE-BR-2015-12; 2014001310009.

## Competing Interests

The authors declare that they have no competing interests.

## Author Contributions

- Jin-Yong Kim conceived and designed the experiments, performed the experiments, analyzed the data, contributed reagents/materials/analysis tools, prepared figures and/or tables, approved the final draft.
- Sanghun Lee analyzed the data, approved the final draft.
- Man-Seok Shin performed the experiments, analyzed the data, contributed reagents/materials/analysis tools, approved the final draft.
- Chang-Hoon Lee performed the experiments, approved the final draft.
- Changwan Seo conceived and designed the experiments, authored or reviewed drafts of the paper, approved the final draft.
- Soo Hyung Eo conceived and designed the experiments, analyzed the data, contributed reagents/materials/analysis tools, authored or reviewed drafts of the paper, approved the final draft.

## Animal Ethics

The following information was supplied relating to ethical approvals (i.e., approving body and any reference numbers):

The ministry of environment provided full approval for this purely observational research.

## Field Study Permissions

The following information was supplied relating to field study approvals (i.e., approving body and any reference numbers):

Field experiments were approved by Institute of Korea National Park Service.

## Data Availability

The raw data have been supplied as a Supplemental File.

## Supplemental Information

Supplemental information for this article can be found online at http://dx.doi.org/10.7717/peerj.4857#supplemental-information.

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
