# Peer review of "Altitudinal patterns in breeding bird species richness and density in relation to climate, habitat heterogeneity, and migration influence in a temperate montane forest (South Korea)"

_PeerJ, doi:10.7717/peerj.4857_

## Round 0.1 · original submission · Major Revisions

Reviewer # 1 remarks:

What I appreciate most about this study is the following an empirical approach. However, clarification of the methods and terminology is required, as well as improvements in language usage. Moreover, this reviewer is also concerned about the mismatch of spatial and temporal resolution of datasets used in the study.

Reviewers 2 and 3 offer constructive comments that will also improve the manuscript upon revision.

Reviewer 1 ·

Basic reporting

This study investigated the breeding bird richness and density distribution along the elevation gradient in 2015 and examined the effects of climate and habitat variables on the observed pattern. What I appreciate most about this study is the following an empirical approach. However, clarification of the methods and terminology is required, as well as improvements in language usage.

Although authors mentioned the widely tested hypothesis in the field in the introduction, I think the introduction can be shortened with a more focused oriented to their own question and hypothesis. Some sentences are long, wordy, and the linkage between following sentences are missing. For instance, Line25-28, Line 28-3. Moreover, the sentences Line 52-54 and Line 55-56 are contrary to each other.

Authors tested the effects of climate and habitat heterogeneity to explain the observed pattern; however, there is no explicitly hypotheses stated in the manuscript. For instance, what do you expect to see is in species richness when understory coverage increases?

The figure captions should give more information about the plots; for instance, you can add one or two sentences to tell what the categories in coverage of vegetation are. Figure axis also should show the unit of the variables such as Fig.3.a, is spring temperature in degrees C?

Please check Table 2. There are two models with delta AICc is equal to 0 for density models.

There is a room for English improvements in the manuscript. There are some unneccary sentences or poor transitions and topic flow between sentences. For instance, Lines 152-156 are unneccary information in this section, becuase neither species turnover or beta diversity is a topic of this manuscript.

Experimental design

My first concern about the mismatch of spatial and temporal resolution of datasets used in the study. The bird surveys were carried out within a group of 50 m radius area, climate variables are downscaled to 1 x 1 km, vertical habitat (vegetation % coverage) is measured in 5m radius areas, and horizontal habitat type and each type area coverage 150 m radius circle. The species observation and vertical habitat heterogeneity datasets covered two months field survey, climate data is gathered from a climate simulation for 3 months, and there is no information for landcover dataset used for calculation horizontal habitat heterogeneity. I strongly recommend adding a clarification in the manuscript to address these spatial and temporal resolution mismatches. An appendix with a graph (scattered or box plot) showing the observed environmental variable among sampling sites would be helpful to understand these patterns as well.

My second concern is the term "migrant rate". I think this phrase can misguide the readers. Instand of migrant rate, authors can consider using "migratory bird ratio". Also please include a justification about using arcsine square root to standardize this data. I think no transformation for this data is required.

Best-fit curves show the relationship between solely one variable and species richness or density. Although Fig.S4 shows only two variables distribution among elevation gradient, a graph showing the altitudinal distribution of each other variable would be really useful to connect these results to the altitudinal gradient. Maybe these graphs can be moved to the main manuscript.

Moreover, I wonder if the authors considered the differences in the number of parameters when comparing models via AICc; for instance, the migrant rate of species (MRS) and spring temperature (SP) looks like a pretending (uninformative) parameters (Table 2), therefore, they should be removed from the models. I recommend the authors reading Arnold (2010)
Arnold, T.W. 2010. Uninformative Parameters and Model Selection Using Akaike’s Information Criterion. J. Wildl. Manage. 74(6): 1175–1178.

Validity of the findings

The manuscript used a dataset from a rarely studied, therefore, it has a benefit to literature. Authors did a good work to connect the ideas in the conclusion to them in the introduction; however, an extensive revision is required to pitch the ideas and deliver these ideas to the readers more clearly.

Reviewer 2 ·

Basic reporting

no comment

Experimental design

no comment

Validity of the findings

no comment

Additional comments

Dear Author:
Your manuscript is interesting, but I have two general remarks:
a) No discussion regarding to the latest literature.
b) It seems to me (unfortunately, based on M & M, I can not evaluation this) that GLM is not a proper analysis, it suggests using GLZ (due to distributions).
Line 4: should be: these still remain
Line 7: please delete „Jirisan National Park”
Line 8: change for „above mean sea level” to „asl”
Line 10: in line 7 you write only about „species richness” now is also density. Please standardize
Line 11: „Models constructed with 16 combinations of” Please, be more precise
Line 13: unclear
Line 14-15: not important for the abstract
Line 17-22: unclear
Line 28: is newest paper about this pattern: Kosicki 2017: Ecological Modeling
Line 30: see paper Kosicki 2017: Environmental modeling and assessment
Line 105: please add size of study plots
Line 110: please write how it was done: LIDAR?
Line 133: I think that the predictors should be extended by: the shape of the patches, area of cover and buffer zone of the patches
Line 161: please extend the results by: relative importance (RI) and evidence ratio. See details in Burnham & Anderson, 2002
Line: 161: correlation and variance inflection factors (VIF) between predictors is necessary,
Line 240: polynomial fucntion ax2+bx+c ?
Line 258: please discuss other aspects of the altitude: roughness, aspect etc.

·

Basic reporting

In my opinion English is clear and professional in all text but I am not native. No information about habitat type in the title. I suggest to change the end of the title “…influence in a montane temperate forest (South Korea)”. The structure of the article is consistent with the journal guidelines. Introduction is sufficient and present a wide context of the study. Study area is too short and it was not described enough. More information is needed about waters, agriculture areas, forests structure and human activity. No information about open areas (e.g. glades) in the study area. The term “above sea level” should be written in abbreviation (a.s.l.) to reduce space. Symbol “]” should be deleted (line 251).The manuscript includes all results relevant to the hypothesis. Results were discussed based on many references. Literature was appropriate referenced. However, reference Burnham KP, Anderson DR, Huyvaert KP (2011) was not cited in the text. Figure 1 is not clear. What are rectangles? Does the line show a border of the Natural Park? In Figure 3e and 4e x axis is described as “Coverage of Overstory vegetaion” and should be “Coverage of Overstory vegetation”. I suggest to move Table 1 to Supplementary files. The worst model should be added (below the best model) for response variables in Table 2. The order should be changed in description of variables in Table 3: first abbreviation and then full description after dash. Some Latin names of species in Table S1 are not correct:
Is: Bubo Bubo; Should be: Bubo bubo
Is: Bonasa bonasia; Should be: Tetrastes bonasia
Is: Troglodyte troglodytes; Should be: Troglodytes troglodytes
Some supplementary files (Fig. S3, Fig. S4 and Table S2) should be placed in Results instead in Discussion.

Experimental design

The study is original research within the aims and scope of the PeerJ journal. Research question is well definied and relevant. However the aims of study should be more general than specific. I suggest to give the type of habitat instead “…at Jirisan National Park, South Korea, situated in a temperate region” (lines 84-85). The study was conducted with high technical and ethical standard. Appropriate permits for study in the Jirisian National Park has been obtained. The research required to gather many variables. Generally methods was sufficiently described. In my opinion statistical methods are appropriate for data analysis. All habitat types used for the habitat diversity index should be presented in Material and Methods (line 146).

Validity of the findings

The presented research is interesting. In my opinion the study provides a new data and broadens a knowledge. The data are sufficient and well analyzed. Conclusion are well stated.

---

## Round 0.2 · accepted · Accept

Congratulations! Your manuscript has been accepted for publication in PeerJ. Note one small change: Table S2 - The name of Tetrastes bonasia was given not correct (is Troglodytes bonasia).

# Reviewer 2 ·

Basic reporting

no coments

Experimental design

no coments

Validity of the findings

no coments

Additional comments

Dear Authors
I agree with your comments and accept your explanations.

·

Basic reporting

In my opinion the manuscript was improved and the new version is better than previous one. The small mistakes must be corrected.

Experimental design

no comment

Validity of the findings

no comment

Additional comments

The manuscript was changed according to the most of suggestions. The better hypotheses were given in introduction. Methods were described clear. New information about habitat was added by inserting new figure (Fig. 2) to methods.
I accept your explanation to my suggestion about changing place of some supplementary files. My intention was changing of the place of citation of these files. I found that table S3 has been cited in results in the manuscript. I agree that Fig. S1 and Fig. S2 are cited in the discussion.

Moreover, I found small mistakes:
Line 13 – I think the better first word of the sentence is „However” than „And”.
Table S2 - The name of Tetrastes bonasia was given not correct (is Troglodytes bonasia).